# The Impact of Different Pedagogical Models on Moderate-to-Vigorous Physical Activity in Physical Education Classes

**DOI:** 10.3390/children9121790

**Published:** 2022-11-22

**Authors:** Élvio Rúbio Gouveia, Jorge Lizandra, Diogo V. Martinho, Cíntia França, Andreas Ihle, Hugo Sarmento, Hélio Antunes, Ana Luísa Correia, Helder Lopes, Adilson Marques

**Affiliations:** 1Department of Physical Education and Sport, University of Madeira, 9020-105 Funchal, Portugal; 2Laboratory of Robotics and Engineering Systems (LARSYS), Interactive Technologies Institute, 9020-105 Funchal, Portugal; 3Center for the Interdisciplinary Study of Gerontology and Vulnerability, University of Geneva, 1205 Geneva, Switzerland; 4Department of Teaching of Physical Education, Artistic and Music of the University of Valencia, 46022 Valencia, Spain; 5University of Coimbra, Research Unit for Sport and Physical Activity (CIDAF), Faculty of Sport Sciences and Physical Education, 3004-504 Coimbra, Portugal; 6Research Center in Sports Sciences, Health Sciences, and Human Development (CIDESD), 5001-801 Vila Real, Portugal; 7Department of Psychology, University of Geneva, 1205 Geneva, Switzerland; 8Swiss National Centre of Competence in Research LIVES—Overcoming Vulnerability: Life Course Perspectives, 1015 Lausanne, Switzerland; 9Interdisciplinary Centre for the Study of Human Performance (CIPER), Faculty of Human Kinetics, University of Lisbon, 1495-751 Lisbon, Portugal; 10Environmental Health Institute (ISAMB), Faculty of Medicine, University of Lisbon, 1649-020 Lisbon, Portugal

**Keywords:** teaching games for understanding, sport education model, instruction model, physical activity, physical education

## Abstract

The aims of this study were twofold: (i) to examine differences in physical activity (PA) variables regarding the length of Physical Education (PE) lessons (45 vs. 90 min) and teaching methodology (Teaching Games for Understanding (TGfU) or hybrid TGfU-sports education (SE)); and (ii) to estimate the percentage of time spent in moderate-to-vigorous (MV) PA assessed via an accelerometer based on total step count, BMI, age, and sex, considering the pedagogical models and the class length. This study was carried out in three classes of invasion games in PE. Two classes were taught based on TGfU (44 students, 18 males, 12.6 ± 0.55), and one was taught based on a hybrid TGfU-SE (17 students, 9 males, 13.7 ± 0.90). The same students were assessed twice, first in a 45 min class and then in a 90 min class. The students’ MVPA was assessed using the Actigraph GT3X Activity Monitor. The results revealed significant differences in PA intensity regarding the length of the PE lesson (45 vs. 90 min) and the teaching methodology. The 45 min classes using TGfU were more intense and had less sedentary time than the 90 min classes. When using a hybrid TGfU-SE, the 90 min classes had higher intensities than the 45 min classes. Generally, when comparing the two types of pedagogical intervention, the TGfU model provided more active lessons and less sedentary time in class than a hybrid TGfU-SE. The TGfU model is more profitable for increasing MVPA in class. Due to the issues related to the organization and management of sports practice in class, several recommendations for maximizing MVPA in a hybrid TGfU-SE class must be considered.

## 1. Introduction

It is well known that high physical activity (PA) and low time devoted to sedentary activities are both wholesome and necessary for children and adolescents [1,2]. Specifically, positive effects have been found at a physiological level, such as weight control and cholesterol levels [3], the development of the musculoskeletal system, and the prevention of cardiovascular diseases [4], as well as social/mental and cognitive benefits in children [5]. In addition, it seems to exert a medium and long-term influence on their lifestyle, helping to maintain the acquired habit in adulthood [6].

Nonetheless, according to recent studies and reports, around 80% of adolescents do not accomplish the recommendations of one daily hour of moderate to vigorous physical activity (MVPA) [1,7], especially girls [8], and spend a large amount of their time in sedentary activities, mainly exposed to screens such as smartphones and tablets, and recreational video gaming or TV viewing [9]. Additionally, the trend observed in most international studies shows a progressive decrease as they approach or reach adulthood [10]. Paradoxically, this happens when there is a great offer of organized sports activities and when physical education (PE) is a subject generally consolidated in educational curriculums.

School, especially PE, is considered an ideal setting for promoting healthy habits [11] since no other institution has such an influence during children’s early years and, consequently, has such an impact on their lifestyles [12]. In this sense, time spent in PE lessons could be, apart from an important indicator of the quality of the school, a determinant context for the fulfillment of PA recommendations, since some studies indicate that effective time spent in a PE class can raise this by 50% [13,14].

Conversely, although interventions in the context of PE and extracurricular activities are frequent, it seems that there is not a sufficient transfer to the active behavior of youths in their leisure time [11,15]. Furthermore, the most appropriate duration of a PE class to guarantee the quantity and quality of PA that produces positive physiological and behavioral changes is not clear [16]. It should also be noted that the features of PE lessons are mediated by the teaching styles and methodologies performed in class. In this sense, previous authors have described greater motor commitment using ludic and freer styles than more directive teaching styles [17]. On the other hand, the authors also concluded that more pragmatic styles reported higher academic performance [18].

Among other factors, the type of PE activity and content are modifiable factors that influence the levels of MVPA in PE [19]. For example, Wang and Wang (2018) confirmed the effectiveness of the Teaching Games for Understanding (TGfU) intervention on the MVPA levels of students from Grades 9 and 10 compared to a traditional technique-based teaching approach [20]. Also, Perlman (2012) found that amotivated students exposed to the Sport Education Model (SEM) were more physically active when compared with the skill–drill–game class [21]. More recently, Melero-Cañas et al. (2021) concluded that a hybrid educational program in PE classes based on Teaching Personal and Social Responsibility and gamification techniques produced enhancements in PA in school [22]. However, few studies have studied the effectiveness of the competence model (i.e., the combination of TGfU and SEM) on the PA in class, considering the length of the PE lesson.

Another interesting indirect indicator, which is very easy to control, is the number of steps, which has reached consistency in recent years [23]. Technological advances have created more reliable and affordable devices for people. Adams et al. (2013) established a correlation among the general recommendations of MVPA for children and adolescents with at least 11,000 steps for an active day [24]. On the other hand, after a 13 country revision, the authors considered that an active session requires a minimum of 100 min (around 5000 steps for a 50 min PE class) [25]. If it is accepted that 50% of the time for a PE class is effective, it could be considered a good strategy for monitoring the intensity of PE lessons to reach 2500 to 3000 steps. In this sense, research supports that working with pedometers would be a good and motivating strategy to implement in PE lessons, as they help observe if students maintain adequate PA levels and control if the class is a good contribution to the total daily PA [26].

Therefore, the aims of this study are twofold. First, to examine differences in PA variables regarding the length of the PE lesson (45 vs. 90 min) and the single teaching methodology, (TGfU) or a hybrid (TGfU-SE). Second, to estimate the percentage of time spent in MVPA assessed via an accelerometer based on step count, BMI, age, and sex, considering the pedagogical models and the class length.

## 2. Materials and Methods

### 2.1. Participants

This study included 61 participants of both sexes (males = 27) aged between 11.8 and 15.6 years old (12.9 ± 0.83). Students participated in the research project entitled “Physical Education in Schools from the Autonomous Region of Madeira” (EFERAM-CIT; https://eferamcit.wixsite.com/eferamcit, accessed on 1 October 2022).). This study was carried out in three classes of invasion games in the PE context in one urban public elementary school in Funchal, Madeira, Portugal. Two classes were taught based on the TGfU model (44 students, 18 males, 12.6 ± 0.55), and one class was taught based on a hybrid TGfU-SE (17 students, 9 males, 13.7 ± 0.90). The same students were assessed twice, first in a 45 min class and then in a 90 min class. All the assessments were performed in the same week. The content taught in these two classes was similar in the two pedagogical approaches (see Table 1). Participants were informed about the study’s objectives and written informed consent was obtained from their legal guardians. The study received ethical approval from the Scientific Committee of the Faculty of Physical Education and Sports at the University of Madeira (Reference: ACTA N.77–12.04.2016). This study was also approved by the Regional Secretary of Education and the school’s headmaster. 

### 2.2. Data Collection Procedure

Moderate to Vigorous Physical Activity (MVPA). The MVPA was assessed using the Actigraph GT3X Activity Monitor. Throughout both didactic units, one lesson of 45 min length and another one of 90 min were selected to be assessed with accelerometers. In total, four classes were fully assessed. In the TGfU ((lesson 6 (45 min) and lessons 7 and 8 (90 min); Table 1), and the hybrid TGfU-SE [(lesson 5 (45 min) and lessons 6 and 7 (90 min), Table 1) all the classes, at this stage of a didactic unit, were teacher-directed instruction. Students were asked to wear the accelerometer on their right hip. The ActiGraph GT3X+ accelerometer was initialized with a 30 Hz sampling frequency. Raw data from gt3x files were converted to 10 s epoch data files prior to analysis. Time spent in MVPA was derived using the ActiLife software, version 6 (ActiGraph, Pensacola, FL, USA), using the cutoff points suggested by the previous literature [27]. The accelerometer was programmed before each lesson, and the data collection started 5 min after the beginning of the class. 

### 2.3. Teaching Models

#### 2.3.1. TGfU Model 

The first group intervention was taught using the TGfU model. The structuring of the teaching content was based on the tactical problems presented in invasion games. Offensive: (i) maintaining possession of the ball; (ii) attacking the goal; (iii) creating and using space in the attack. Defensive: (i) defending space, (ii) defending the goal; (iii) winning the ball [28]. In each of these tactical problems, individual and collective tactical-technical skills (i.e., off-the-ball movements and on-the-ball skills) were developed throughout modified games (i.e., small side games). The planning of exercises considering the level of game complexity is described by Mitchel et al. (2013) [28]. The teaching and training progression started with small-sized games (2 vs. 2 players, 3 vs. 3 players), progressing to a maximum of 6 players per team at the advanced level. Further explanation of the rationale for this teaching approach and a detailed description of the exercise components are reported elsewhere [28]. This didactic unit comprises an initial assessment (90 min), then 26 lessons (990 min), and the final assessment (90 min). The whole didactic unit comprises 1170 min (19.5 h), with 8 × 45 min and 9 × 90 min classes.

#### 2.3.2. Hybrid TGfU-SE 

The second group was taught using a hybrid TGfU-SE (i.e., TGfU + SEM; [29]). All the intervention characteristics described for the TGfU Model above were also applied to this group intervention. This didactic unit comprises two-moment assessments (i.e., initial and final 135 min), then the pre-season (5 lessons, 360 min; 2 × 45 min and 3 × 90 min classes), season, (8 lessons, 585 min; 3 × 45 min and 5 × 90 min classes), and the culminating event–festivity (1 lesson, 90 min). The whole didactic unit was composed of 1170 min (19.5 h). Further explanation of the rationale for this teaching approach and a detailed description of the class components and organization are mentioned elsewhere [30]. The instruction steps of the two groups are shown in Table 1. Both interventions were administered by three trainee PE teachers, supervised by two experienced PE teachers. The training PE teachers involved have more than three years of experience in invasion games training outside of the school context.

### 2.4. Data Analysis

Descriptive statistics for Sedentary time, Light PA, Moderate PA, Vigorous PA, Very Vigorous PA, and MVPA in minutes and percentages, and the total step counts, as well as the average step counts per minute, were calculated separately for TGfU and hybrid TGfU/SE and 45 min and 90 min classes. Second, a Wilcoxon signed-rank test was conducted to examine if there was a change in all the variables studied in the same students between 45 min (moment 1) and 90 min (moment 2) classes, considering different pedagogical models (TGfU and hybrid TGfU-SE). Secondly, a mixed between-within-subjects analysis of variance was conducted to assess the impact of the two different pedagogical models (TGfU and hybrid TGfU-SE) on MVPA (%), Sedentary Time (%), and the Average of Steps by minute (n) across the two classes (45 and 90 min). Data analysis was performed using IBM SPSS v26 (IBM Corp., Armonk, NY, USA) and Graphpad Prism (version 5.00 for Windows, GraphPad Software, San Diego, California, USA, www.graphpad.com, accessed on 2 October 2022). The significance level was set at *p* < 0.05. 

## 3. Results

Table 2 shows all PA intensities in terms of time and percentage, the total number of steps, and the average number of steps per minute for the 45 min and 90 min classes following different pedagogical models (TGfU and hybrid TGfU-SE). 

In the classes of TGfU, a Wilcoxon signed-rank test showed significantly higher time in Light PA, Moderate PA, Vigorous PA, MVPA, and average step counts in the 45 min compared to 90 min classes. Opposite results were seen for sedentary time and total step counts, where the 90 min classes showed significantly higher values than the 45 min classes. Similar results were seen for the hybrid TGfU-SE, except on Sedentary (%), Very Vigorous (%), and MVPA (%). 

A mixed between-within-subjects analysis of variance was conducted to assess the impact of the two different pedagogical models (TGfU and hybrid TGfU-SE) on participants’ MVPA (%) across the two classes (45 and 90 min). 

There was a significant interaction between the pedagogical intervention model and the length of the class: Wilks’ Lambda = 0.74, F (1, 59) = 21.33, *p* < 0.001, partial eta squared = 0.27. There was a substantial main effect for classes: Wilks’ Lambda = 0.85, F (1, 59) = 10.75, *p* < 0.002, partial eta squared = 0.15, with the TGfU model showing a reduction in MVPA from 45 to 90 min classes. The opposite results were seen in the hybrid TGfU-SE, showing an increase in MVPA from 45 to 90 min classes (Figure 1).

The main effect comparing the two types of pedagogical intervention was significant, F (1, 59) = 21,33, *p* < 0.001, partial eta squared = 0.27, suggesting differences between the two teaching approaches across different length of class, with the TGfU model providing a more active lesson than hybrid TGfU-SE.

In sedentary time, there was also a significant interaction between the pedagogical intervention model and the length class: Wilks’ Lambda = 0.76, F (1, 59) = 19.02, *p* < 0.001, partial eta squared = 0.24. There was a substantial main effect for classes, Wilks’ Lambda = 0.83, F (1, 59) = 12.04, *p* = 0.001, partial eta squared = 0.17, with the TGfU model showing an increase in sedentary time from 45 to 90 min classes, and the hybrid TGfU-SE reducing the sedentary time from 45 to 90 min classes (Figure 2). The main effect comparing the two types of pedagogical intervention was significant, F (1, 59) = 20,15, *p* < 0.001, partial eta squared = 0.26, suggesting differences between the two teaching approaches across different length classes, with the TGfU model providing a less sedentary lesson than hybrid TGfU-SE. 

Finally, in average step counts per minute, there was also a significant interaction between the pedagogical intervention models and the length of class: Wilks’ Lambda = 0.81, F (1, 55) = 13.10, *p* = 0.001, partial eta squared = 0.19. There was a substantial main effect for classes, Wilks’ Lambda = 0.81, F (1, 55) = 12.85, *p* = 0.001, partial eta squared = 0.19, with the TGfU model showing a significant decrease in average step count per minute from 45 to 90 min classes (Figure 3). The main effect comparing the two types of pedagogical intervention was significant, F (1, 55) = 9,89, *p* = 0.001, partial eta squared = 0.19, suggesting differences between the two teaching approaches across different length of class, with the TGfU model showing again as a more active lesson than hybrid TGfU-SE.

### 3.1. Regression Equation Using Step Count, Body Mass Index, Age, and Sex to Estimate the MVPA (%) in Class for TGfU and Hybrid TGfU-SE Considering the Class Length 

To estimate the percentage of time spent in MVPA assessed via an accelerometer in class, a multiple regression analysis was performed, with step count, BMI, age, and sex as the main predictors. Considering the two different pedagogical models and class length, four regression equations were developed. 

### 3.2. TGfU Model

#### 3.2.1. 45 min Classes Using TGfU Model

Y = 48.35 + 0.018(X) + 0.006(Y) + [(−1.89)(Z)], + 1.56 (Sex), where X = number of steps; Y = BMI; Z = age, Sex (0 = girl and 1 = boy); R^2^ = 0.83; SEE = 4.1; Durbin–Watson = 1.86. The number of steps (β = 0.87; *p* < 0.001) was a significant predictor.

#### 3.2.2. 90 min Classes Using TGfU Model

Y = 60.96 + 0.009 (X) + 0.053(Y) + [(−2.00)(Z)], + [(−4.05)(Sex)], where X = number of steps; Y =BMI; Z= age, Sex (0 = girl and 1 = boy); R^2^ = 0.47; SEE = 6.8; Durbin–Watson = 1.74. The number of steps (β = 0.54; *p* = 0.001) was a significant predictor.

### 3.3. Hybrid TGfU-SE Model

#### 3.3.1. 45 min Classes Using Hybrid TGfU-SE

Y = 6.12 + 0.027 (X) + (−0.394)(Y) + (0.437)(Z), + 0.653 (Sex), where X = number of steps; Y =BMI; Z= age, Sex (0 = girl and 1 = boy); R^2^ = 0.82; SEE = 4.1; Durbin–Watson = 2.28. The number of steps (β = 0.97; *p* < 0.001) was a significant predictor.

#### 3.3.2. 90 min Classes Using Hybrid TGfU-SE

Y = 10.50 + 0.004 (X) + 0.34(Y) + 1.68(Z), + [(−4.35)(Sex)], where X = number of steps; Y =BMI; Z= age, Sex (0 = girl and 1 = boy); R^2^ = 0.28; SEE = 8.18; Durbin–Watson = 1.63.

## 4. Discussion

This study aimed to examine differences in PA variables regarding the length of the PE lesson (45 vs. 90 min) and the single teaching methodology (TGfU) or a hybrid (TGfU-SE). Also, based on the step count, BMI, age, and sex, regression equations were developed to estimate the percentage of time spent in MVPA assessed via an accelerometer, considering the pedagogical models and the class length. The 45 min classes using TGfU are more intense and have less sedentary time than the 90 min classes. When using the hybrid TGfU-SE, the 90 min classes have higher intensities than the 45 min classes. In both teaching approaches, the 90 min lessons have twice the number of steps compared to the 45 min lessons, as expected. However, when considering the correction, i.e., the average step count per minute metric, we found higher step counts in 45 min classes. Generally, when comparing the two types of pedagogical intervention, the TGfU model provided more active lessons (i.e., MVPA and Steps) and less sedentary time in class than the hybrid TGfU-SE. This study’s first message is that the TGfU model’s use in teaching invasion sports is preferable in shorter classes compared to longer classes. Teaching Games for Understanding is a game-centric model that focuses on a student-centered approach, exploring the teaching of tactics and techniques in a game context [31]. The strategy is to play small-sided games in the sessions, improve the experience and learning environment, and enhance the development of in-game skills. Several studies have proven that TGfU-based interventions increase PA levels. They can help achieve the recommended MVPA time in PE classes (50% class time) [20,32]. In the present study, in 45 and in 90 min classes, we achieved 59.8% and 50.1% of MVPA attained, which goes forward with those recommendations. The main reason given is related to the nature of the games. The small-side team games seem to enhance the students’ experience of freedom and enjoyment during the games, and that could justify the high MVPA levels observed in the TGfU classes. Also, it has been recognized that students who are taught using the TGfU model are found to be more autonomous and critical thinkers [33], and consequently, they are more engaged.

Concerning the strategic objectives of SEM, these are to make a more cultured student (i.e., to know about the sports phenomenon) more competent (i.e., the development of skills to play a game) and more enthusiastic (i.e., to engage in roles beyond being a team player, contributing actively for an in-class positive sporting atmosphere) [30]. The current results support the idea that when the main focus is to increase the MVPA in class, using hybrid TGfU-SE is not the best strategy. It is believed that the high time dedicated to roles other than that of student player [34] decreases the PA intensity of the class. Indeed, the class time dedicated to the organization and management of the sports practice by the students themselves may justify the lower MVPA seen in the hybrid TGfU/SE intervention compared to the single TGfU intervention. In agreement, previous research concluded that combining TGfU and SE teaching is more labor-intensive because teachers have to possess superior content and pedagogical content knowledge [35].

The previous literature has focused on the hybrid TGfU-SE intervention in PE classes, mainly as a successful way to positively affect students’ motivation or social development [36,37,38]. However, few studies have examined the efficacy of a hybrid TGfU-SE unit to attain the recommended MVPA time in PE classes. Studies comparing a single TGfU pedagogical intervention to a hybrid TGfU-SE pedagogical one are lacking. This is an important issue, because TGfU and SE are two well-known curriculum models physical educators use to provide sporting experiences in the PE context. Secondly, this study supports that we need longer classes to reach higher PA intensities when using SEM-based hybrid models. This is an important practical recommendation for physical educators to accrue high levels of MVPA while delivering a desirable curriculum model for social growth and responsibility. Apart from this recommendation, Pennington (2019) also suggested a couple of recommendations for maximizing the SEM; namely, limiting the time spent giving instruction and management, encouraging a faster pace during transitions and gameplay, using simple and well-known small-sided games, and increasing training on how to address the need for MVPA adequately. Physical educators must consider these important recommendations when using SEM to contribute to accomplishing the recommendations of weekly MVPA. 

The third message of the present study concerns the duration of the classes. Teaching Games for Understanding enhances the intensity of PA in the class and reduces sedentary time compared to a hybrid TGfU-SE. This issue is particularly important since increasing PA levels in PE classes is a priority strategy for achieving PA for health guidelines [39,40]. On the other hand, a recent systematic review and meta-analysis of moderate-to-vigorous PA levels in secondary school PE lessons [41] underlines the importance of looking for new research and intervention strategies to increase classroom activity time. This review showed in-class MVPA values of 40.5%, below the values of our study and previously reported by the US Center for Disease Control and Prevention (2010) and the UK Association for Physical Education recommendation of 50% [39]. The present study seeks to point to the use of teaching models that help to enhance MVPA in the classroom. The current results highlight the importance of using teaching models to improve students’ learning experiences, such as TGfU. Indeed, the key concept of this teaching model is the design of appropriate games (i.e., all students must be successful) that allow students to understand the principles of the game and, at the same time, increase intrinsic motivation [42]. Following this approach, students will find participation in games more interesting, motivating, and authentic than skill-building exercises that have little application in the game [43]. 

Finally, this study reinforces the importance of monitoring the intensity of PE classes to potentiate class time to achieve weekly PA recommendations. The absence of effective measures to assess the intensity of PE classes in the classroom has been a limitation pointed out previously [41]. For this reason, our study presents specific equations to help PE educators to assess the intensity of shorter classes (45 min) or long classes (90 min) using the single TGfU or a hybrid TGfU-SE. Thus, based on the number of steps taken in class, BMI, age, and sex, which are now easy measures to assess in the classroom context, teachers and educators now have equations that help them to monitor the intensity of their classes, contributing to a large extent to students’ awareness of the minimum level they have to achieve in class.

Some limitations of the present study should be recognized. First, the small sample size and the unbalanced samples across the different teaching methodologies. Future studies must consider more extensive samples and a similar number of participants in each intervention group. To overcome this limitation, participants were assessed twice in the same week. Also, two expert PE teachers supervised the intervention, and the protocol was fulfilled in full. Second, the reduced number of classes assessed: four classes, two for TGfU and the other for the hybrid TGfU-SE approach. This limitation is partly smoothed by using accelerometers to assess PA, which provides an objective and valid assessment. Finally, one of this study’s biggest strengths is providing practical information that allows PE teachers and educators to monitor the PA intensity of their classes. Considering the importance of increasing the MVPA in class, teachers should consider this information to have more accurate information about the impact of PE on the weekly total PA.

## 5. Conclusions

This study gives practical indications about how physical educators could potentiate the time spent in MVPA in class, considering the length of the class and the instructional teaching model (i.e., TGfU or a hybrid TGfU-SE). It is preferable to use the TGfU approach in lower-length classes instead of a hybrid TGfU-SE. Also, considering only the TGfU model, this approach is more profitable to increase MVPA in 45 min than 90 min classes. Due to the issues related to the organization and management of sports practice in class, several recommendations for maximizing the SEM must be considered when considering a hybrid TGfU-SE approach.

## Figures and Tables

**Figure 1 children-09-01790-f001:**
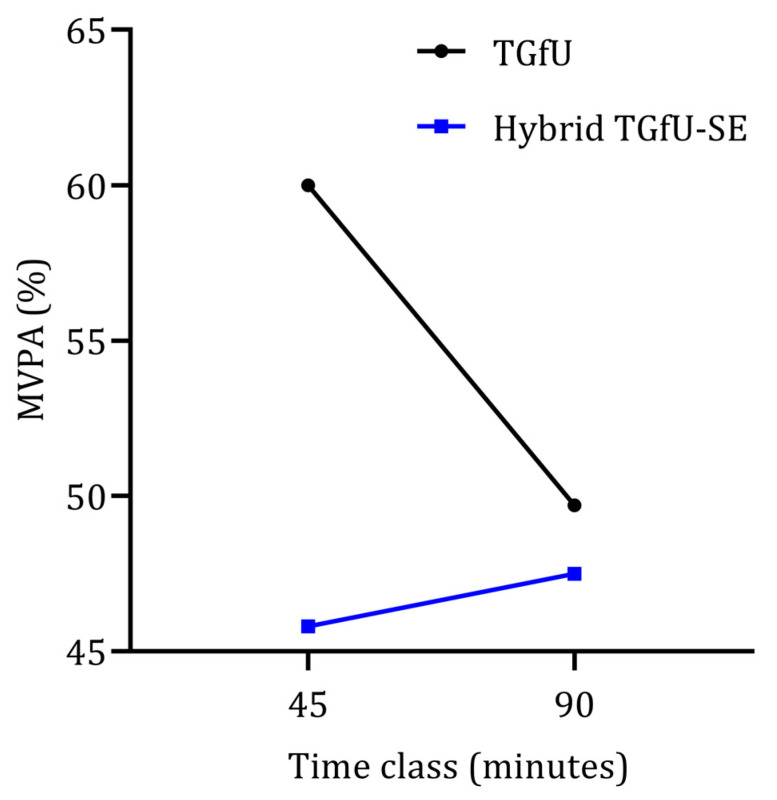
MVPA variation from 45 min to 90 min classes considering two different pedagogical models (TGfU and hybrid TGfU-SE).

**Figure 2 children-09-01790-f002:**
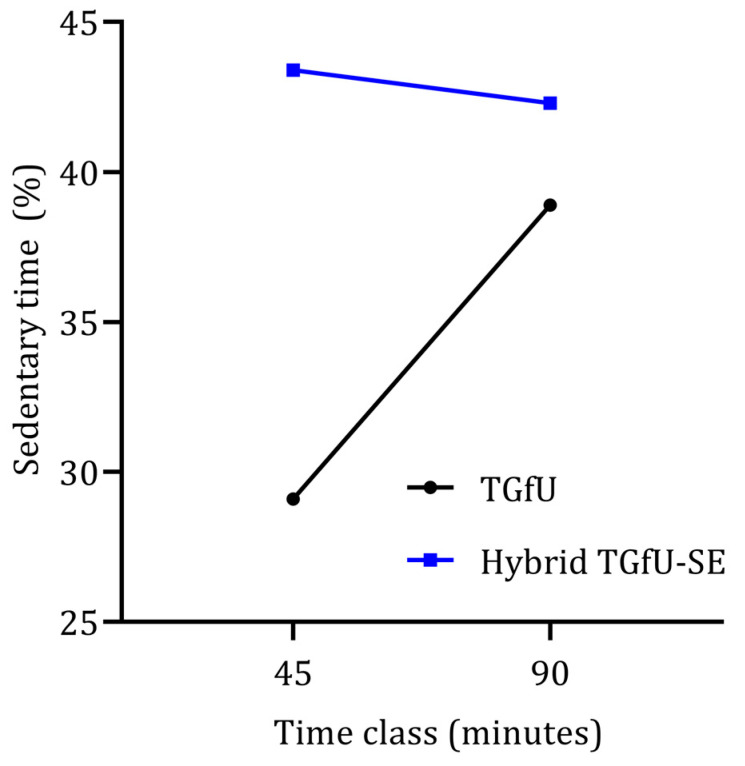
Sedentary time variation from 45 min to 90 min classes considering two different pedagogical models (TGfU and hybrid TGfU-SE).

**Figure 3 children-09-01790-f003:**
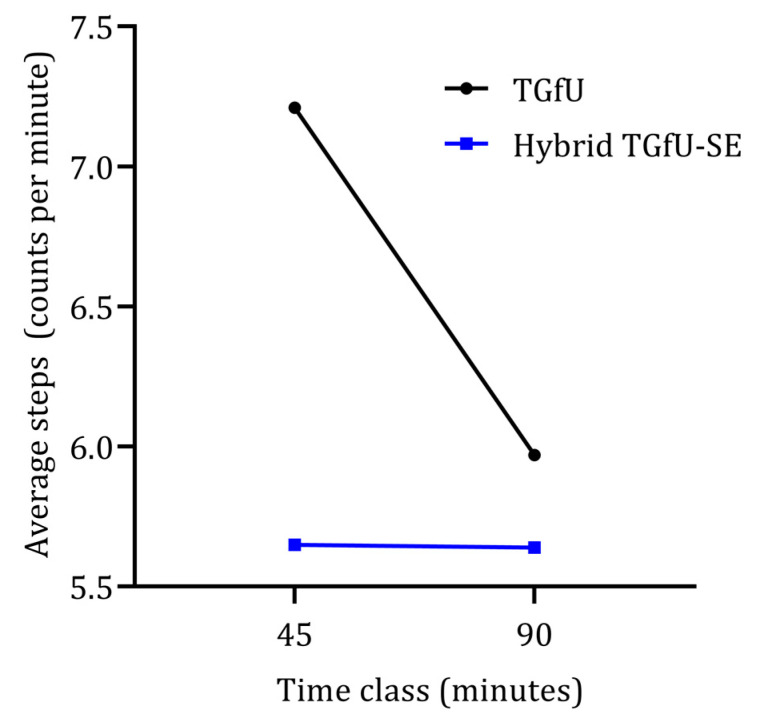
Average step count per minute variation from 45 min to 90 min classes considering two different pedagogical models (TGfU and hybrid TGfU-SE).

**Table 1 children-09-01790-t001:** Season plan for the hybrid TGfU-SE and TGfU units.

Lesson	Duration (min)	TGfU (Learning Contents)	Lesson	Duration (min)	Hibrid TGfU-SE (Learning Contents)
1 and 2	90	Initial assessment	1	45	Initial assessment
3	45	Maintaining the possession of the ball/Defending space	2	45	Initial assessment
4 and 5	90	Maintaining the possession of the ball/Defending space	3 and 4	90	Teacher-directed instruction: Introduction to the concept of the season. Explanation of the model and competition format. Learning situations for Maintaining the possession of the ball / Defending space
6	45	Maintaining the possession of the ball/Defending space	5	45	Teacher-directed instruction: Learning situations for Maintaining the possession of the ball / Defending space
7 and 8	90	Maintaining the possession of the ball/Defending spaceAttacking the goal /Defending the goal	6 and 7	90	Teacher-directed instruction: Learning situations for Attacking the goal /Defending the goal
9	45	Attacking the goal /Defending the goal	8	45	Teacher-directed instruction: Learning situations for Creating and Using space in attack/Winning the ball
10 and 11	90	Attacking the goal /Defending the goal	9 and 10	90	Teacher-directed instruction: Learning situations for Creating and Using space in attack/Winning the ball; Teacher-directed instruction within team practice: Final remarks
12	45	Attacking the goal /Defending the goal	11	45	Student-directed instruction: instruction within team practice.
13 and 14	90	Maintaining the possession of the ball/Defending space/Attacking the goal/Defending the goal	12 and 13	90	Student-directed instruction: 1º Championships for season points
15	45	Maintaining the possession of the ball/Defending space/Attacking the goal/Defending the goal	14 and 15	90	Student-directed instruction within team practice.
16 and 17	90	Maintaining the possession of the ball/Defending space/Attacking the goal/Defending the goal	16 and 17	90	Student-directed instruction within team practice. Student-directed instruction: 2º Championships for season points
18	45	Maintaining the possession of the ball/Defending space/Attacking the goal/Defending the goal	18	45	Student-directed instruction: 2º Championships for season points
19 and 20	90	Creating and Using space in attack/Winning the ball	19 and 20	90	Student-directed instruction within team practice
21	45	Creating and Using space in attack/Winning the ball	21	45	Student-directed instruction within team practice
22 and 23	90	Creating and Using space in attack/Winning the ball	22 and 23	90	Student-directed instruction: 3º Championships for season points
24	45	Creating and Using space in attack/Winning the ball	24	45	Final Assessment
25 and 26	90	Final Assessment	25 and 26	90	Culminating event–Festivity

**Table 2 children-09-01790-t002:** PA intensities, the number of total steps (n), and the average step count per minute for 45 min and 90 min classes following different pedagogical models (TGfU and hybrid TGfU-SE).

	TGfU Model					Hybrid TGfU-SE				
	45 min		90 min			45 min		90 min		
	Mean	SD	Mean	SD	*p*	Mean	SD	Mean	SD	p ^†^
Sedentary (min)	13.25	3.92	34.79	6.65	<0.001	19.55	4.32	38.09	5.08	<0.001
Light (min)	4.84	1.13	10.06	2.01	<0.001	4.85	1.76	9.15	2.16	<0.001
Moderate (min)	20.73	3.07	34.81	5.60	<0.001	14.73	2.56	32.60	3.42	<0.001
Vigorous (min)	5.26	2.35	9.17	3.54	<0.001	5.39	2.17	9.22	3.24	<0.001
Very Vigorous (min)	0.92	0.72	1.17	1.08	0.223	0.48	0.60	0.94	0.97	0.025
MVPA (min)	26.91	4.23	45.14	6.61	<0.001	20.60	3.70	42.76	4.67	<0.001
Sedentary (%)	29.44	8.71	38.66	7.39	<0.001	43.44	9.60	42.33	5.65	0.423
Light (%)	10.76	2.50	11.18	2.23	0.175	10.78	3.92	10.16	2.40	0.513
Moderate (%)	46.08	6.81	38.67	6.22	<0.001	32.72	5.68	36.23	3.80	0.007
Vigorous (%)	11.68	5.23	10.18	3.93	0.005	11.98	4.83	10.24	3.60	0.053
Very Vigorous (%)	2.04	1.60	1.30	1.19	<0.001	1.07	1.33	1.04	1.08	0.887
MVPA (%)	59.81	9.40	50.16	7.35	<0.001	45.77	8.22	47.51	5.18	0.196
Steps Count (*n*)	1947.74	459.91	3225.56	517.72	<0.001	1529.06	293.01	3048.50	477.12	<0.001
Avg Step Count (*n*)	7.21	1.71	5.97	.96	<0.001	5.65	1.09	5.64	0.87	<0.001

† Wilcoxon signed-rank test.

## Data Availability

The data presented in this study are available upon request from the corresponding author.

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
