# Peer review of "The Impact of Different Pedagogical Models on Moderate-to-Vigorous Physical Activity in Physical Education Classes"

_children, 2022, doi:10.3390/children9121790_

Round 1

Reviewer 1 Report

Dear authors, thank you for this interesting study. There are only a few comments to revise in your manuscript, exspecially in results and discussion/ conclusion.

Please see table 2: You present MVPA and sedentary times only in %. Please add these values in minutes. The percentages do not reach 100%, if I add MVPA and sedentary. What about the rest of the time? What is this time used for? Please add this information. Steps are presented in n. So, a significant difference between 45 min and 90 min PE ist a logic conclusion. What about to present steps per min or steps per MVPA? Would this show a significant difference? Please add these data.

Figures: Please add information in the legends about abbreviations. Teaching models are there described as TGfU and MEC, in the legend ist TGfU and TGfU-SE.

Regression anaysis: i do not understand the sense here to have only the n of steps as predictor and no other variable?

Discussion: Please add at first in one sentence the aim of the study. 

About lines 290 and following, these would be conclusions in my opinion.

Please correct typos (e.g. TGFU vs. TGfU) and grammar in the whole text.

Author Response

Reviewer 1

Dear authors, thank you for this interesting study. There are only a few comments to revise in your manuscript, especially in results and discussion/ conclusion.

Dear reviewer, thanks for your positive approach to this paper. Your comments on the manuscript are too pertinent and will improve the quality.

1 - Please see table 2: You present MVPA and sedentary times only in %. Please add these values in minutes. The percentages do not reach 100%, if I add MVPA and sedentary. What about the rest of the time? What is this time used for? Please add this information. Steps are presented in n. So, a significant difference between 45 min and 90 min PE ist a logic conclusion. What about to present steps per min or steps per MVPA? Would this show a significant difference? Please add these data.

Answer 1: The reviewer is correct on all points. As suggested, we added all the information requested. First, we added the complete information about these classes: Sedentary, Light PA, Moderate PA, Vigorous PA, Very Vigorous PA, and MVPA. Second, we also integrate a new metric: Average step counts per minute. 

2 - Figures: Please add information in the legends about abbreviations. Teaching models are there described as TGfU and MEC, in the legend ist TGfU and TGfU-SE.

Answer 2: the reviewer is right. The legend in the figures was corrected. 

3 - Regression anaysis: i do not understand the sense here to have only the n of steps as predictor and no other variable?

Answer 3: Our idea of putting all one predictor, the number of steps, was to build a simple equation regression that is easy to use in practice by the physical educators using only one variable that today is easy to get from different tolls. However, after this comment, we integrated BMI, age, and sex. We believe this information is easy to get and improves the MVPA prediction.  

4 - Discussion: Please add at first in one sentence the aim of the study. 

Answer 4: As suggested by the reviewer, the aim of this study was introduced in the first paragraph of the discussion.

5 - About lines 290 and following, these would be conclusions in my opinion.

Answer 5: the idea here was to reinforce the main results of this study, making the message stronger to physical educators.

6 - Please correct typos (e.g. TGFU vs. TGfU) and grammar in the whole text.

Answer 6: we have corrected TGFU throughout the text. A revision of the grammar was done. 

Reviewer 2 Report

Please check name of red line in Figure 1,

if it is a "hybrid TGfU-SE"?

Author Response

Reviewer 2

Dear reviewer, thanks for your positive approach to this paper. As suggested, the legend in the figures was corrected.